# Advancing Quantization Steps Estimation : A Two-Stream Network Approach for Enhancing Robustness

Xin Cheng *
Nanjing University of Information Science and Technology
Nanjing, China
chengxin0314@126.com

Hao Wang *
Huaiyin Institute of Technology
Huai'an, China
sa875923372@163.com

Jinwei Wang †
Nankai University
Tianjin, China
wjwei_2004@163.com

Xiangyang Luo†
State Key Laboratory of Mathematical Engineering and Advanced Computing
Zhengzhou, China
luoxy_ieu@sina.com

Bin Ma
Qilu University of Technology
Jinan, China
sddxmb@126.com

## Abstract

In Joint Photographic Experts Group (JPEG) image steganalysis and forensics, the quantization step can reveal the history of image operations. Several methods for estimating the quantization step have been proposed by researchers. However, existing algorithms fail to account for robustness, which limits the application of these algorithms. To solve the above problems, we propose a two-stream network structure based on Swin Transformer. The spatial domain features of JPEG images exhibit strong robustness but low accuracy. Conversely, frequency domain features demonstrate high accuracy but weak robustness. Therefore, we design a two-stream network with the multi-scale feature of Swin Transformer to extract spatial domain features with high robustness and frequency domain features with high accuracy, respectively. Furthermore, to adaptively fuse features in both the frequency domain and spatial domain, we design a Spatial-frequency Information Dynamic Fusion (SIDF) module to dynamically allocate weights. Finally, we modify the network from a regression model to a classification model to speed up convergence and improve the accuracy of the algorithm. The experiment results show that the accuracy of the proposed method is higher than 98% on clean images. Meanwhile, in robust environments, the algorithm proposed maintains an average accuracy of over 81%.

## CCS Concepts

• **Applied computing** → *Investigation techniques*; • **Computing methodologies** → *Computer vision*.

---

*Both authors contributed equally to this research.
†The corresponding author.

---

## Keywords

Image forensics, JPEG compression, Robust quantization step estimation, Swin Transformer Network

**ACM Reference Format:**
Xin Cheng, Hao Wang, Jinwei Wang, Xiangyang Luo, and Bin Ma. 2024. Advancing Quantization Steps Estimation : A Two-Stream Network Approach for Enhancing Robustness. In *Proceedings of the 32nd ACM International Conference on Multimedia (MM '24), October 28-November 1, 2024, Melbourne, VIC, Australia.* ACM, New York, NY, USA, 9 pages. https://doi.org/10.1145/3664647.3680736

## 1 Introduction

Images serve as an essential medium for conveying information in the lives of people. They have an irreplaceable role in communication, education, and entertainment. However, with the maturity of image editing software, the authenticity of an image cannot be determined solely by the naked eye. Tampered images are often used by unscrupulous individuals to mislead public perception, thereby generating negative impacts on society. Consequently, image forensics has attracted extensive attention from governments and judicial authorities. Due to its less storage space and faster transmission speed, the JPEG format is widely popular in real life [32]. Therefore, researchers have carried out many research works on JPEG images. Examples include JPEG steganography [5, 19, 36, 41], JPEG steganalysis [12, 20, 26, 40], double JPEG compression forensics [22, 30, 31], and JPEG image quantization step estimation [3, 6, 9–11, 13–15, 17, 18, 21, 23, 25, 28, 29, 33–35, 37–39].

In the field of JPEG forensics, if we can accurately estimate the quantization step, we can then get the compression information of the original JPEG image. This serves as a crucial basis for determining whether an image has been tampered with or contains concealed information. For example, in image steganography, the quantization step can provide edge information to correct the steganographic cost and improve security. In steganalysis, existing algorithms tend to be trained for a specific quantization step, which leads to cover source mismatch easily. If researchers can accurately estimate the quantization step, they can design appropriate models for different quality factors (QF) to solve the above problem. In image forensics

and localization, Niu *et al.* [24] segmented the image into several patches. Then, they determined the tampered regions by estimating the quantization step of each patch and attributed multiple tampered regions based on the inconsistency of the quantization matrix.

Therefore, the quantization step has become an important parameter in image steganography and image forensics. For a single compressed JPEG image, we can obtain information such as the quantization step from the header file. However, once an image is tampered with, it will inevitably be re-saved. Whether stored in lossless BMP format or lossy JPEG format, the quantization step of the first compression will be overwritten. Therefore, specific algorithms need to be designed to estimate the quantization step. The existing methods for estimating the quantisation step can be divided into traditional methods and deep learning-based methods. Traditional methods estimate the quantization step based on the distribution characteristics of Discrete Cosine Transform (DCT) coefficients. However, in robust environments, noise can alter the distribution of DCT histograms, leading to lower accuracy in traditional algorithms.

With the development of deep learning, researchers have proposed some neural network-based methods [6, 23, 29] to estimate the quantization step. Compared to traditional methods that use information from fixed locations to estimate the quantization step. The neural network-based methods use a convolutional kernel to mine the correlation between different locations as auxiliary information to estimate the quantization step. Therefore, these methods perform well on robust environment. The methods in [23] and [29] estimate the quantization step based on spatial domain information. However, the image information loss during JPEG compression is mainly due to quantization errors and truncation/rounding errors, which occur in the frequency domain [6]. Therefore, the frequency domain features tend to contain rich compression traces. The method in [6] estimates the quantization step by mining the information in the frequency domain, so its performance is better than [23] and [29].

However, current methods only consider robustness under double compression and do not take into account the interference of other noises, such as Gaussian filtering, rotation, cropping, triple compression, and so on. To address these issues, we conduct an analysis of spatial domain information and frequency domain information.

Although the frequency domain contains more compression traces, this feature is more susceptible to noise interference. Therefore, it exhibits higher accuracy in clean samples, while its accuracy is lower in noisy images. The spatial domain information, on the other hand, is obtained by overlapping the frequency domain information, and thus it is a dispersed compression trace. Therefore, this compression trace is less vulnerable to noise interference, exhibiting stronger robustness. To balance the accuracy and robustness simultaneously, we propose a two-stream network model that allows the two branches of the network to extract compression traces of the image in both the frequency and spatial domains. Furthermore, considering the differences between spatial and frequency domain features, a simple superimposition may result in the omission and conflation of features. As a result, we propose the spatial-frequency domain information fusion module to assign weights to these two

features adaptively. Finally, we analyze more deeply the impact of regression and classification models on the accuracy of estimating quantization steps in a robust environment. Compared to the regression model, the classification model has faster convergence and higher accuracy. Consequently, we design an end-to-end classification model and use the cross-entropy loss function to measure the difference between the estimated and true quantization steps.

The main contributions of this paper can be summarized as:

- We propose a two-stream network based on the Swin Transformer model. The frequency domain features represent deep-level characteristics but exhibit lower robustness. On the other hand, spatial domain features represent shallow-level characteristics but possess stronger robustness. Therefore, we separately extract these two features to design a highly accurate and robust network.
- We propose a Spatial-frequency Information Dynamic Fusion (SIDF) module based on the attention mechanism. By dynamically allocating weights to the characteristics of the frequency and spatial domains, the network can achieve a adaptive feature fusion effect.
- We conduct a more in-depth analysis of the differences between classification and regression models in robust environments. Subsequently, we devise an end-to-end classification model to speed up the convergence of the network and further enhance its accuracy.

## 2 Related works

Based on the characteristics of JPEG dequantization, the DCT coefficients, after dequantization, are distributed at integer multiples of the quantization step. Luo *et al.* [18], after analyzing the quantization error as well as rounding/truncation errors in JPEG, deduced that the DCT coefficients do not align precisely at integer multiples of the quantization step. Instead, they exhibit a higher distribution probability within the range of [-1, +1] multiples of the quantization step. Based on the above findings, Luo *et al.* [18] deduced that the position where the DCT histogram reaches its peak corresponds to the quantization step. Building upon Luo *et al.* [18] work, Yang *et al.* [35] further decomposed the DCT coefficient histogram and devised statistical measures for a factor histogram. They selected the histogram block with the highest index from histograms, surpassing a certain threshold as the quantization step for each frequency. Ye *et al.* [38] estimated the quantization step by computing the power spectrum of the DCT coefficient histogram. Lin *et al.* [15] proposed a content-adaptive algorithm that estimates the quantization step based on the Energy Density Spectrum (EDS) of the DCT coefficient histogram and the Fourier transform of the EDS.

When dealing with small-sized images, the accuracy of the quantization step estimate tends to degrade rapidly due to the insufficient DCT coefficients available. To address this issue, Yang *et al.* [34] proposed a clustering-based framework to enhance the accuracy of existing methods. The method involved merging frequency information with the same quantization step through clustering, and then estimating the quantization step using the merged DCT coefficients. Li *et al.* [14], by observing the strong correlation between the unique shape of DCT coefficient distribution and quantization steps, designed a function that generates candidate step with similar

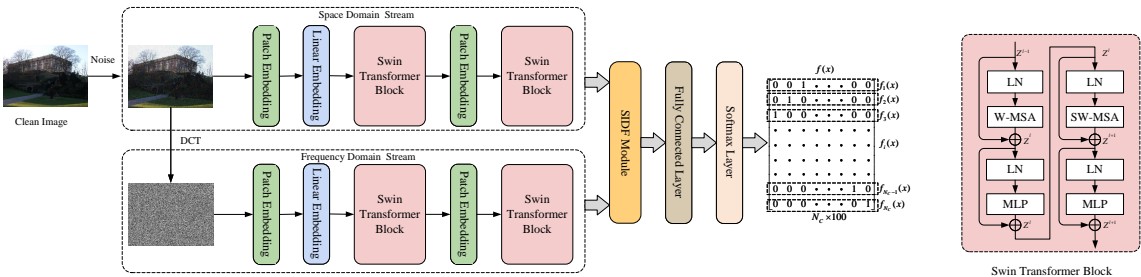

**Figure 1: Overview of the proposed two-stream network. 1) Extracting features in the spatial and frequency domains by using a two-stream network, respectively. 2) Using the SIDF-module to dynamically assign weights to spatial domain features and frequency domain features. 3) Designing a classification architecture to improve the accuracy of the algorithm.**

shapes. They analyzed this function to estimate the quantization step based on this observation.

For recompressed images, Bianchi and Piva [3] proposed modeling the distribution of DCT coefficients at a specific frequency as a mixture model of modified and unchanged components. They utilized the mixed model to construct a maximum likelihood estimation function, and then employed the expectation maximum (EM) algorithm to determine the first quantization step for each frequency. In response to the split noise and residual noise generated by secondary compression, Galvan *et al.* [11] devised a filtering strategy to alleviate the interference caused by these two types of errors. They also developed an error function to determine candidate quantization steps.

With the rise of deep learning, Niu *et al.* [23] were the first to introduce a method utilizing a convolutional neural network to estimate the first quantization step. They trained the network as a standard regression problem to estimate the quantization step. Building upon Niu *et al.* [23] work, Tondi *et al.* [29] simultaneously considered accuracy and mean squared error in constructing the loss function. Additionally, they introduced a network structure resembling classification methods. Cheng *et al.* [6] designed a preprocessing in the frequency domain to cluster DCT coefficients at the same frequency. Simultaneously, they introduced the Res2Net-C network architecture, which employs hierarchical connections within residual blocks to capture multi-scale features in images.

## 3 The Proposed Method

In this section, we will detail the specific aspects of estimating the quantization step through a two-stream network. As depicted in Fig. 1, we first introduce the two-stream network model designed based on the Swin Transformer architecture. Subsequently, an attention mechanism-based space-frequency information dynamic fusion module is proposed. Finally, an end-to-end classification structure network is proposed to speed up the convergence of the network and improve the accuracy of the algorithm.

### 3.1 Two-stream network based on the Swin Transformer model

By analyzing the compression process of JPEG, it can be observed that the image information loss during JPEG compression is mainly due to quantization errors and truncation/rounding errors, which

occur in the frequency domain [6]. Therefore, compared to spatial domain information, compression artifacts in the frequency domain are more pronounced and more accessible to acquire. However, when an image is subjected to noise interference, compression artifacts in the frequency domain are also more susceptible to disruption. Therefore, using frequency domain information for feature extraction achieves excellent performance on clean images, but its effectiveness is poor on noisy images.

On the other hand, the information in the spatial domain is a stacking of the information in the frequency domain, which contains insignificant traces of compression. Hence, algorithms relying on spatial domain information exhibit lower accuracy on clean images than those utilizing frequency domain information. However, due to the more concealed nature of its compression artifacts, this results in a lesser degree of disruption caused by noise, thereby contributing to the stronger robustness of these algorithms. To balance the robustness and accuracy of the algorithm, we design a two-stream network. The two-streams of the network extract information from the spatial and frequency domains of the image, respectively.

In addition, the network models of existing methods are based on traditional CNN [6, 23, 29]. Recently, the Transformer model has achieved significant success in directions such as image classification [8, 16] and object detection [4]. We first attempt to design a network model based on the Transformer model. Cheng *et al.* [6] have pointed out that multi-scale features in images contribute to extracting compression artifacts. Swin Transformer introduces a hierarchical pathway mechanism that enables information propagation across different levels, thereby obtaining multi-scale information from images. Therefore, we design the network for estimating the quantization step based on the Swin Transformer network.

As shown in Fig. 1, each stream utilizes two Swin Transformer blocks, and each block consists of two submodules. Each submodule comprises a LayerNorm (LN) layer, an attention module, followed by another LN layer, and a multi-layer perceptron (MLP) layer. The first submodule employs the window multi-head self-attention (W-MSA) module, while the second submodule utilizes the shifted window multi-head self-attention (SW-MSA) module. The shift operation between these two submodules facilitates information exchange among windows. For estimating quantization steps, the MSA module allows the network to simultaneously focus

Xin Cheng, Hao Wang, Jinwei Wang, Xiangyang Luo, and Bin Ma

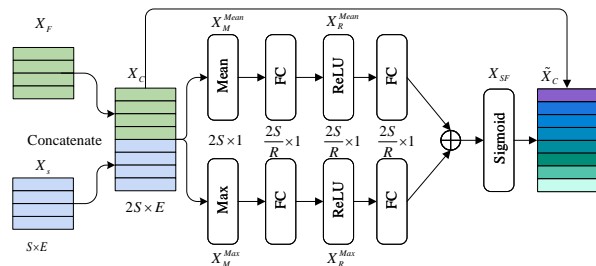

Figure 2: Space-frequency domain feature dynamic fusion module.

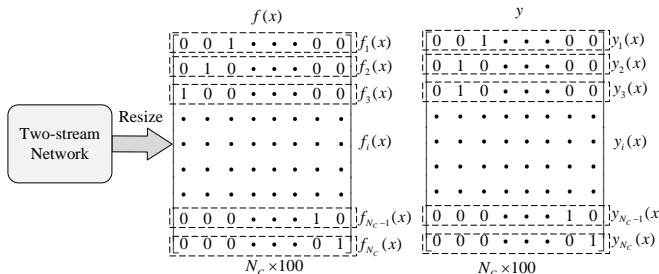

Figure 3: A end-to-end classification architecture network.

on compression artifacts across different dimensions. Additionally, employing self-attention within each head enables the network to allocate varying weights based on the complexity of image textures. This capability allows the network to extract more compression artifacts from regions with high texture complexity.

Moreover, the hierarchical architecture within the network enables it to learn multi-scale information from images. The compression process in JPEG typically introduces compression traces and block artifacts, which become more pronounced at different scales. By extracting multi-scale features, the model can better capture these artifacts. Finally, local shifts are allowed between windows in Swin transformer, which allows the network to capture both local and global information. The performance of the algorithm is improved by combining local and global compression traces.

## 3.2 Space-Frequency Information Dynamic Fusion (SIDF) Module

We have obtained image spatial domain and frequency domain features using two-stream network, respectively. However, the frequency domain features are easily extracted shallow features, while the spatial domain represents deep features that are harder to extract. Therefore, a simple superimposition may result in the omission and conflation of features.

We propose a SIDF-Module to solve the above problem. This module combines the attention mechanism to dynamically allocate weights to the spatial and frequency domain features. As shown in Fig. 2, the input of the two-stream network is a spatial domain feature map $X_S$ and a frequency domain feature map $X_F$. The size of the feature map is $S \times E$, where $S$ represents the spatial dimensions of the feature map, and $E$ represents the feature dimension at each spatial position. The process of generating an attention mechanism for the feature map is as follows:

First, we perform dimension expansion by merging the feature maps $X_S$ and $X_F$ into $X_C$.

$$X_C = [X_S, X_F] \tag{1}$$

Then, we employ two functions, computing the average and the maximum values, to squeeze the spatial positional feature dimension $E$.

$$X_M^{\text{Mean}} = \frac{1}{E} \sum_{j=1}^{E} X_C(i, j) \tag{2}$$

$$X_M^{\text{Max}} = \max_i X_C(i, j) \tag{3}$$

Following that, we employ two Fully Connected (FC) layers to reduce and expand the spatial dimensions, applying the ReLU function directly for non-linear mapping in both operations. Furthermore, the features from the two branches are summed element-wise and then passed through a Sigmoid activation function.

$$X_{SF} = \text{Sigmoid}\left(X_R^{\text{Max}} + X_R^{\text{Mean}}\right) \tag{4}$$

Finally, we use residual connection to obtain the fused feature map $\tilde{X}_C$ and assign weights to the spatial features.

$$\tilde{X}_C = X_C + X_{SF} \tag{5}$$

## 3.3 A end-to-end classification architecture network

Most traditional methods for estimating the quantization step are based on the distributional characteristics for the histogram of the DCT coefficients. However, if the image undergoes noise processing, the distribution of DCT coefficients will be altered. This implies that traditional methods have poor robustness. By adding noise to the inputs of the network, it helps the network learn more robust and generalized features. This approach helps to reduce overfitting to the training data and increases the ability of the network to resist interference.

The existing neural network-based methods for estimating the quantization step can be divided into regression model-based methods [6, 23] and classification model-based methods [29]. The papers [2, 29] pointed out that for the estimation of discrete numbers, the commonly used approach involves employing the softmax function followed by the cross-entropy function, which aids in the process of backpropagation. Therefore, we further explore the network structure for classification in the environment of adding noise.

As shown in Fig. 3, we design a neural network model with a classification structure. In this architecture, each quantization step $q_i$ is encoded as a one-hot vector. We use the more common quantization step for compression, i.e., $q_i \leq 100$. Therefore, for a given quantization step, the length of its one-hot code is set to $1 \times 100$. As shown in Fig. 3, the output of the network after cross-entropy is resized to a matrix $f(x)$ of size $N_C \times 100$. Here, $N_C$ represents the number of expected estimated quantization steps. The row vector $f_i(x)$ represents the one-hot encoded estimated

**Table 1: The performance of comparative experiment on single compress images.**

| $QF_1$ | UCID | | | | | | | RAISE | | | | | | |
|---|---|---|---|---|---|---|---|---|---|---|---|---|---|---|
| | Luo [18] | Yang [35] | Li [14] | Niu [23] | Tondi [29] | Cheng [6] | Ours | Luo [18] | Yang [35] | Li [14] | Niu [23] | Tondi [29] | Cheng [6] | Ours |
| 60 | 43.53 | 42.55 | 55.47 | 30.11 | 96.61 | 96.14 | **99.35** | 32.28 | 27.65 | 34.52 | 29.73 | 88.55 | 88.62 | **96.55** |
| 65 | 47.80 | 46.04 | 56.58 | 37.19 | 94.00 | 95.53 | **98.37** | 34.66 | 28.53 | 39.23 | 24.16 | 86.59 | **90.99** | 90.70 |
| 70 | 52.57 | 50.68 | 59.30 | 38.47 | 96.35 | 97.33 | **98.11** | 37.00 | 33.44 | 41.65 | 25.07 | 90.63 | 91.57 | **92.42** |
| 75 | 57.63 | 57.49 | 64.30 | 43.19 | 96.65 | 97.84 | **99.30** | 38.98 | 35.58 | 43.69 | 29.61 | 92.23 | 91.92 | **95.64** |
| 80 | 64.09 | 65.47 | 72.07 | 45.43 | 98.09 | 98.39 | **98.60** | 44.74 | 43.19 | 49.14 | 30.76 | 91.82 | 94.47 | **95.65** |
| 85 | 68.01 | 69.30 | 78.06 | 49.27 | 98.40 | 97.23 | **99.60** | 50.74 | 57.04 | 58.55 | 32.66 | 92.64 | 95.02 | **96.08** |
| 90 | 72.00 | 75.26 | 85.66 | 71.24 | 98.69 | **99.13** | 98.54 | 56.12 | 57.04 | 66.26 | 36.47 | 91.30 | 95.89 | **97.19** |
| 95 | 72.07 | 90.22 | 89.79 | 91.07 | 99.16 | **99.94** | 99.65 | 63.34 | 64.53 | 72.52 | 53.76 | 95.21 | 97.78 | **98.38** |
| Average | 59.71 | 62.12 | 70.15 | 50.74 | 97.24 | 97.69 | **98.94** | 44.73 | 43.37 | 50.69 | 32.77 | 91.12 | 93.28 | **95.32** |

quantization steps, and $y_i(x)$ represents the one-hot encoded true quantization steps. The formula for calculating the quantization step by $f_i(x)$ and $y_i(x)$ is as follows:

$$\widehat{q}_i = \arg\max_j f_{ij}(x), i = 1, 2, \ldots, N_C, j = 1, 2, \ldots, 100 \quad (6)$$

$$q_i = \arg\max_j y_{ij}(x), i = 1, 2, \ldots, N_C, j = 1, 2, \ldots, 100 \quad (7)$$

Here, $\widehat{q}_i$ represents the estimated quantization steps and $q_i$ represents the true quantization steps. We set the loss function as the cross entropy loss, formulated as follows:

$$\mathcal{L}(x) = \frac{1}{N_C} \sum_{i=1}^{N_C} \left( \sum_{j=1}^{100} y_{ij} \log \left( f_{ij}(x) \right) \right) \quad (8)$$

In this formulation, we first calculate the cross-entropy loss for each quantization step and then average all the losses.

## 4 Experiments

### 4.1 Experimental Settings

**Datasets**: To evaluate our method, we conducted experiments on three challenging datasets: Uncompressed Color Image Database (UCID) [27], Raw Images Dataset (RAISE) [7] and Natural Resources Conservation Service (NRCS) [1]. We selected 900 uncompressed images from database. Considering the realistic applications of quantization step in tamper detection and localization, we randomly cropped each image into 10 patches of size $64 \times 64$. Consequently, we obtained a total of 9, 000 patches. Subsequently, we compressed these patches using quality factors $QF_1 = \{60, 65, 70, 75, 80, 85, 90, 95\}$ and stored them in JPEG format. After a single compression 9, 000 × 8 patches were obtained. We divided the training set, validation set, and testing set in a ratio of 6 : 2 : 2. Therefore, a total of $4.32 \times 10^4$ patches were used for training, $1.44 \times 10^4$ patches were used for validation and $1.44 \times 10^4$ patches were used for testing. Finally, we employ MATLAB for random processing of images. These operations include the following five: **Gaussian noise** with a mean value of 0 and variance ranging from 1 to 8 at intervals of 1. **Gaussian filtering** with filter kernel of 3 and variance ranging from 0.1 to 0.8 at intervals of 0.1. **Double compression** with $QF_2$ from 65 to 100 at intervals of 5. **Triple compression** with $QF_2$ and $QF_3$ from 80 to 100 at intervals of 10. **Rotation attacks** with angle from 90 to 360 at intervals of 90.

**Implementation details:** We use Adam optimizer with a learning rate of $10^{-3}$. The batch size for training and testing is set to 80.

we train 80 epochs on the network and select the minimum loss epochs from the validation set as the best network.

### 4.2 Performance Comparison with Other Schemes

To investigate the ideal performance of the proposed algorithm in the absence of noise attacks, we perform comparative experiments on a single compressed image. As shown in Table 1, we select three deep learning methods for comparison, including, Niu *et al.* [23], Tondi *et al.* [29] and Cheng *et al.* [6]. Besides, we also choose three traditional methods for comparison, including Luo *et al.* [18], Yang *et al.* [35] and Li *et al.* [14].

Table 1 shows that the proposed method has the best performance, and the accuracy on the UCID dataset is more than 98% in all cases. Compared with the traditional method, the proposed method is 28% higher than the method in [14], 36% higher than the method in [35], and 39% higher than the method in [18]. The reason is that the methods in [18] and [35] estimate the quantization step based on the distributional properties of DCT coefficients histograms. However, the method in [35] further decomposes the DCT coefficients to construct the factor histograms, and thus its accuracy is higher than the method in [18] on high-quality images. The method in [14] represents the distribution of DCT coefficients using mathematical functions, exhibiting higher accuracy on high-quality images. However, in small-sized low-quality images, where the DCT coefficients of the image are compressed within the [-1, 1] range, the effective information decreases. Consequently, its accuracy is low under conditions of low-quality images.

Several neural network-based methods have been proposed to improve the accuracy of low-quality images. These methods extract traces of the compression process to estimate the quantization step, using convolutional kernels to extract relationships between different frequencies as auxiliary information. Therefore, they can address the issue of insufficient effective information in low-quality images that traditional algorithms encounter. In these neural network-based methods, the proposed method is 1% higher than the method in [6], 2% higher than the method in [29], and 47% higher than the method in [23]. The reason is that the methods in [23] and [29] estimate the quantization step by extracting the spatial domain information. However, JPEG information loss occurs mainly in the frequency domain, so information in the spatial domain contains fewer compression traces. Therefore, the accuracy of the spatial domain-based method is lower than that of the frequency domain-based method [6]. The method in [29] improves

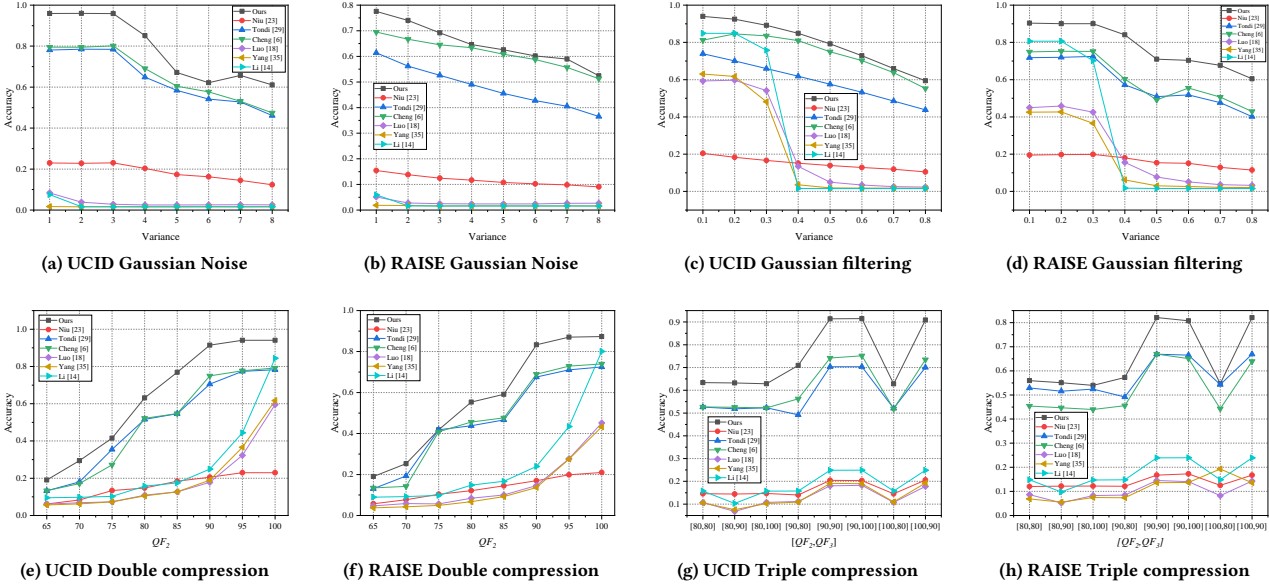

(a) UCID Gaussian Noise     (b) RAISE Gaussian Noise     (c) UCID Gaussian filtering     (d) RAISE Gaussian filtering

(e) UCID Double compression     (f) RAISE Double compression     (g) UCID Triple compression     (h) RAISE Triple compression

**Figure 4: Robustness against different noise attacks compared with[6, 14, 18, 23, 29, 35].**

the loss function based on the method in [23] and further improves the ability to extract features, so its accuracy is higher than that of [23].The proposed method extracts information in both spatial and frequency domains and further optimizes the loss function. Therefore, it has the highest accuracy rate.

## 4.3 Robustness Comparison with Other Schemes

For robustness testing, we add a noise layer to the proposed method and the deep learning-based algorithms[6, 23, 29] during training to ensure experimental fairness.

**Robustness of the Proposed Method Against Image Gaussian Noise Attacks**: Gaussian noise and Gaussian filtering are commonly used methods for assessing robustness. In Fig. 4a and Fig. 4b, as the variance increases, the accuracy of all algorithms decreases. This is due to higher variance leading to increased noise intensity, which has a more significant impact on the image. Consequently, the accuracy of the algorithms shows a declining trend. The accuracy of all three traditional methods in the case of Gaussian noise is less than 10%, which indicates that the traditional methods are unable to resist the Gaussian noise attack. The reason is that the traditional algorithm mainly estimates the quantization step based on the distribution characteristics of the image DCT coefficient histogram. If Gaussian filtering or Gaussian noise is applied to the image, it will affect the distribution of DCT, thus impacting the accuracy of the algorithm.

On the other hand, deep learning-based methods extract the compression trace of a whole image, which is less affected by Gaussian filtering and Gaussian noise. Among the three neural network based algorithms, the method in [6] based on frequency domain features is higher than the methods in [23, 29] based on spatial domain features. This indicates that the frequency domain features are more

useful for the network to extract compression traces. However, the proposed two-stream network architecture of this approach integrates spatial domain information as auxiliary features on the basis of frequency domain information. It leverages the SIDF-module to harmonize information from both domains. Consequently, the proposed method exhibits the best performance.

**Robustness of the Proposed Method Against Image Gaussian Filtering Attacks**: As shown in Fig. 4c and Fig. 4d, similar to Gaussian noise, the accuracy of these algorithms decreases as the intensity of the noise filtering increases. The proposed method has the highest accuracy and is still greater than 60% when the intensity of Gaussian filtering is highest. It is worth noting that the traditional algorithm still has good performance at $variance \leq 0.3$. However, when $variance > 0.3$, the accuracy of the traditional algorithm drops to 20%. The reason for this is that when $variance \leq 0.3$, the Gaussian filtering fails to affect the distribution of the histogram of the DCT coefficients, thus the algorithms remain effective. However, when $variance > 0.3$, the Gaussian filtering causes the histogram of the DCT distribution to be severely shifted, thus making these algorithms ineffective.

**Robustness of the Proposed Method Against Image JPEG compression Attacks**: As shown in Fig. 4e and Fig. 4f, we conduct a second compression using $QF_2 = [65, 70, 75, 80, 85, 90, 95, 100]$ to test the robustness of the algorithm against JPEG compression attacks. As can be seen in Fig. 4e and Fig. 4f, the accuracy of these algorithms increases as the quality factor increases. The reason for this is that the second compression will cover the traces of the first compression. The larger the quality factor, the lesser the compression of the image, the more features are available to estimate the quantization step of the first compression. Consequently, this leads to an increase in accuracy. To further explore the robustness of the algorithm under multiple compression scenarios, as depicted in

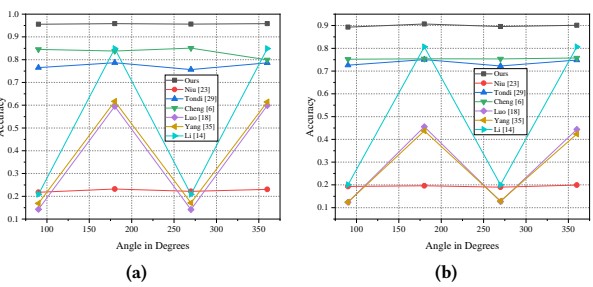

**Figure 5: Robustness against rotation attacks compared with[6, 14, 18, 23, 29, 35]. (a) UCID database. (b) RAISE datebase**

Fig. 4g and Fig. 4h, we conducted three compressions on the image using quality factors $QF_2 = [80, 90, 100]$ and $QF_3 = [80, 90, 100]$. In this scenario, not only does the second compression cover the traces of the first compression, but the third compression also masks the traces of the first compression, thereby significantly increasing the difficulty in estimating quantization steps. As can be seen in Fig. 4g and Fig. 4h, the accuracy is lower after triple compression than after secondary compression. However, the proposed method achieves the highest accuracy for both secondary compression and triple compression.

**Table 2: The performance of generalization experiments with training on UCID, testing at RAISE and NRCS**

| | RAISE (Gaussian Noise) | | | | NRCS (Gaussian Noise) | | | |
|---|---|---|---|---|---|---|---|---|
| $Variance$ | Niu [23] | Tondi [29] | Cheng [6] | Ours | Niu [23] | Tondi [29] | Cheng [6] | Ours |
| 1 | 15.67 | 60.87 | 68.15 | **76.01** | 18.47 | 69.34 | 77.80 | **87.93** |
| 2 | 14.14 | 56.89 | 68.11 | **72.10** | 17.13 | 65.05 | 78.99 | **85.56** |
| 3 | 12.90 | 53.66 | 66.59 | **69.09** | 15.57 | 60.89 | 77.20 | **81.83** |
| 4 | 12.02 | 50.56 | 65.66 | **66.27** | 14.63 | 58.07 | 74.86 | **78.32** |
| 5 | 11.23 | 46.92 | **63.04** | 62.64 | 13.42 | 53.48 | 71.27 | **73.51** |
| 6 | 10.31 | 44.27 | **59.82** | 58.51 | 12.65 | 50.19 | 67.50 | **67.94** |
| 7 | 9.47 | 41.37 | **56.88** | 53.50 | 11.73 | 46.63 | **61.86** | 61.83 |
| 8 | 8.82 | 37.33 | **51.78** | 47.14 | 10.52 | 43.06 | **57.36** | 55.36 |
| | RAISE (Gaussian Filtering) | | | | NRCS (Gaussian Filtering) | | | |
| $Variance$ | Niu [23] | Tondi [29] | Cheng [6] | Ours | Niu [23] | Tondi [29] | Cheng [6] | Ours |
| 0.1 | 18.88 | 69.36 | 72.87 | **83.80** | 21.17 | 76.66 | 76.96 | **92.09** |
| 0.2 | 19.08 | 69.29 | 72.78 | **83.88** | 20.91 | 76.49 | 76.92 | **91.85** |
| 0.3 | 18.80 | 68.86 | 72.93 | **83.47** | 21.41 | 76.28 | 77.18 | **91.87** |
| 0.4 | 16.59 | 53.57 | 58.72 | **70.14** | 18.65 | 63.45 | 66.28 | **80.78** |
| 0.5 | 14.61 | 49.06 | 52.09 | **59.14** | 16.63 | 58.04 | 58.53 | **66.06** |
| 0.6 | 14.45 | 49.92 | 55.41 | **56.98** | 14.85 | 55.42 | 57.55 | **60.45** |
| 0.7 | 12.97 | 45.65 | 50.35 | **57.97** | 12.92 | 51.58 | 53.29 | **61.42** |
| 0.8 | 11.50 | 39.28 | 44.07 | **50.72** | 11.72 | 47.35 | 48.39 | **59.06** |
| | RAISE (Double Compression) | | | | NRCS (Double Compression) | | | |
| $QF_2$ | Niu [23] | Tondi [29] | Cheng [6] | Ours | Niu [23] | Tondi [29] | Cheng [6] | Ours |
| 65 | 5.85 | 12.09 | 12.04 | **17.49** | 5.93 | 13.44 | 13.22 | **18.40** |
| 70 | 7.33 | 16.44 | 14.99 | **24.23** | 7.87 | 19.47 | 17.85 | **28.40** |
| 75 | 11.58 | 37.39 | **34.32** | 33.91 | 12.69 | 30.87 | 27.17 | **38.98** |
| 80 | 12.79 | 42.10 | 42.72 | **52.12** | 14.12 | 42.32 | 50.63 | **60.27** |
| 85 | 15.32 | 44.07 | 56.01 | **64.27** | 16.71 | 46.87 | 52.92 | **70.10** |
| 90 | 17.57 | 65.55 | 67.80 | **74.75** | 19.33 | 69.47 | 71.03 | **86.39** |
| 95 | 18.43 | 71.19 | 71.28 | **79.80** | 20.50 | 75.55 | 75.65 | **89.43** |
| 100 | 18.46 | 71.03 | 72.39 | **81.23** | 21.00 | 76.27 | 78.07 | **89.33** |
| | RAISE (Triple Compression) | | | | NRCS (Triple Compression) | | | |
| $[QF_2, QF_3]$ | Niu [23] | Tondi [29] | Cheng [6] | Ours | Niu [23] | Tondi [29] | Cheng [6] | Ours |
| [80, 80] | 13.24 | 50.42 | 45.01 | **52.63** | 14.13 | 52.78 | 52.58 | **59.32** |
| [80, 90] | 13.31 | 49.95 | 43.84 | **52.59** | 14.05 | 52.98 | 51.80 | **59.32** |
| [80, 100] | 12.91 | 49.62 | 44.36 | **52.14** | 14.39 | 51.89 | 52.12 | **59.62** |
| [90, 80] | 12.72 | 48.14 | 44.18 | **53.73** | 13.35 | 50.02 | 52.65 | **66.56** |
| [90, 90] | 17.31 | 65.12 | 62.34 | **73.83** | 19.12 | 69.30 | 70.54 | **85.84** |
| [90, 100] | 17.02 | 65.76 | 64.58 | **75.63** | 18.92 | 68.72 | 71.83 | **85.84** |
| [100, 80] | 13.11 | 50.25 | 43.82 | **51.39** | 13.86 | 52.10 | 51.27 | **58.65** |
| [100, 90] | 17.31 | 65.12 | 62.34 | **73.83** | 19.12 | 69.30 | 70.54 | **85.84** |
| | RAISE (Rotation) | | | | NRCS (Rotation) | | | |
| $Angle$ | Niu [23] | Tondi [29] | Cheng [6] | Ours | Niu [23] | Tondi [29] | Cheng [6] | Ours |
| 90 | 18.98 | 67.80 | 71.91 | **81.78** | 19.91 | 71.18 | 79.82 | **90.98** |
| 180 | 18.54 | 72.71 | 73.17 | **84.12** | 20.89 | 77.04 | 80.95 | **92.12** |
| 270 | 19.23 | 67.98 | 72.24 | **81.63** | 20.35 | 71.56 | 79.92 | **91.46** |
| 360 | 18.81 | 73.05 | 69.43 | **83.57** | 21.17 | 77.61 | 76.59 | **92.24** |

**Robustness of the Proposed Method Against Image Rotation Attacks**: Image rotation is a common geometric attack. As shown in Fig. 5, we rotate the image with $angle = [90, 180, 270, 360]$.

From Fig. 5, it can be observed that traditional algorithms fail at rotation angles of [90, 270] degrees while remaining effective at angles of [180, 360]. It is strongly influenced by the angle of rotation. However, the deep learning based methods have good performance for different rotation angles. In which, the accuracy of the proposed method is greater than 90%, which is 10% higher than the method in [6], 17% higher than the method in [29], and 72% higher than the method in [23].

## 4.4 Generalization experiments

To test the generalizability of the algorithm, Table 2 shows the results of training with the UCID and testing with RAISE and NRCS . From Table 2, it can be seen that the proposed method achieves the best performance in all cases. Taking the RAISE dataset as an example and comparing Fig. 4 with Table 2, we observe that the proposed algorithm exhibits no significant decrease in accuracy when applied across different databases. In fact, in certain instances of noise attacks, its accuracy even improves. The reason for this is that although the information in the spatial domain varies from different databases. However, the frequency domain is an enriched information of the spatial domain, so the difference of information in the frequency domain between different databases is not significant. The proposed method simultaneously extracts the information in the frequency and spatial domains, and then uses a SIDF-module to dynamically fuse the space-frequency domain features. Therefore, the proposed algorithm is also less affected by database differences and has good generalization.

## 4.5 Ablation study

**The number of Swin Transformer blocks**: We choose one, two and three Swin Transformer blocks for the experiments respectively, and the experiment results are shown in Table 3. From Table 3, it can be seen that the accuracy is highest using two blocks, and the accuracy of three blocks is lower than that of two blocks, and one block has the lowest accuracy. The reason is that the network can learn deeper features as the number of blocks increases. Therefore, the accuracy of two blocks is higher than one layer block. However, downsampling in the Swin Transformer network architecture is performed before each Patch Merging to reduce resolution. This results in a fourfold reduction in the size of the feature map. Our proposed network takes an input size of $64 \times 64$. After one block, the feature map size becomes $32 \times 32$. After two blocks, the feature map size becomes $16 \times 16$. After three blocks, the feature map size becomes $8 \times 8$. Excessively small feature map sizes can cause the neural network to lose crucial compressed information, resulting in lower accuracy for three blocks than two blocks.

**SIDF module**: To test the performance of the SIDF module, we train the network with and without the SIDF module, respectively. As can be seen in Table 3, we compare the performance of the proposed SIDF module for different databases. The use of the SIDF module results in the best performance in all cases and an average accuracy of at least 2% higher than without the SIDF module. The reason is that spatial domain information represents a relatively shallow yet robust feature, while frequency domain information constitutes a deeper and less robust feature. The SIDF module dynamically integrates spatial and frequency domain information,

**Table 3: The ablation study on the number of Swin Transformer blocks and SIDF modules.**

| $QF_1$ | UCID | | | | | | RAISE | | | | | | NRCS | | | | | |
|---|---|---|---|---|---|---|---|---|---|---|---|---|---|---|---|---|---|---|
| | one block | | two blocks | | three blocks | | one block | | two blocks | | three blocks | | one block | | two blocks | | three blocks | |
| | ✓ | × | ✓ | × | ✓ | × | ✓ | × | ✓ | × | ✓ | × | ✓ | × | ✓ | × | ✓ | × |
| 60 | 80.34 | 78.65 | **84.86** | 83.25 | 77.53 | 75.71 | 69.61 | 62.85 | **82.41** | 77.79 | 78.38 | 74.01 | 80.35 | 74.40 | **83.38** | 81.08 | 82.43 | 81.15 |
| 65 | 55.54 | 55.38 | 77.37 | 75.97 | **77.53** | 68.91 | 58.56 | 54.35 | **69.28** | 60.72 | 59.27 | 49.39 | 50.16 | 51.3 | 60.08 | 57.71 | **61.50** | 55.13 |
| 70 | 75.13 | 69.63 | **81.73** | 79.99 | 79.51 | 78.64 | 70.38 | 70.29 | 71.49 | 70.18 | **74.43** | 63.83 | 65.83 | 64.84 | **82.23** | 66.27 | 68.91 | 71.40 |
| 75 | 59.39 | 60.57 | **80.43** | 73.16 | 75.56 | 74.59 | 65.89 | 61.22 | **75.32** | 69.77 | 69.16 | 64.40 | 60.72 | 54.27 | **78.78** | 71.92 | 73.26 | 66.45 |
| 80 | 74.35 | 63.79 | **81.94** | 80.56 | 79.88 | 76.26 | 63.70 | 60.30 | **75.77** | 74.51 | 67.70 | 67.36 | 61.84 | 58.47 | **78.52** | 72.60 | 74.09 | 71.95 |
| 85 | 60.26 | 60.86 | **84.37** | 78.84 | 82.59 | 80.51 | 57.00 | 56.66 | **73.86** | 70.87 | 70.31 | 64.61 | 58.53 | 56.94 | **79.95** | 74.24 | 73.92 | 70.42 |
| 90 | 66.42 | 63.51 | **84.70** | 80.42 | 82.75 | 79.96 | 66.54 | 53.68 | **71.28** | 70.63 | 69.38 | 70.27 | 63.44 | 73.70 | **77.72** | 73.21 | 74.62 | 74.27 |
| 95 | 70.30 | 67.64 | **77.96** | 77.95 | 77.55 | 75.13 | 61.63 | 60.01 | **74.05** | 72.56 | 72.96 | 68.70 | 67.16 | 61.82 | **79.54** | 66.89 | 65.88 | 64.69 |
| Average | 67.71 | 65.00 | **81.67** | 78.76 | 79.11 | 76.21 | 64.16 | 59.92 | **74.18** | 70.87 | 70.19 | 65.32 | 63.50 | 61.96 | **77.52** | 70.49 | 71.82 | 69.43 |

The ✓ represents the use of the SIDF module, while the × indicates the non-use of the SIDF module.

achieving optimal fusion effects. Therefore, the accuracy of the SIDF module is higher than that of the non-SIDF module.

**Table 4: Performance of different branches of the network.**

| $QF_1$ | Clean image | | | Noisy image | | |
|---|---|---|---|---|---|---|
| | *Sp* | *Fre* | *Sp&Fre* | *Sp* | *Fre* | *Sp&Fre* |
| 60 | 96.25 | 96.68 | **99.35** | 83.77 | 82.74 | **84.86** |
| 65 | 96.83 | 97.65 | **98.37** | 72.15 | 72.90 | **77.37** |
| 70 | 95.67 | 96.20 | **98.11** | 78.82 | 76.79 | **81.73** |
| 75 | 97.98 | 99.10 | **99.30** | 79.80 | 79.11 | **80.43** |
| 80 | 96.96 | 98.20 | **98.60** | 79.87 | 79.18 | **81.94** |
| 85 | 97.25 | 98.20 | **99.60** | 75.65 | 74.30 | **84.37** |
| 90 | 97.30 | 98.45 | **98.54** | 82.12 | 79.51 | **84.70** |
| 95 | 98.80 | 99.30 | **99.65** | 70.32 | 70.35 | **77.96** |
| Average | 97.13 | 97.97 | **98.94** | 77.81 | 76.86 | **81.67** |

**The number of branches in the network**: As shown in Table 4, we test the performance of using spatial domain stream, using frequency domain stream and fusing spatial frequency domain stream, respectively. Where, *Sp* represents the use of the space stream, *Fre* represents the use of the frequency stream and *Sp&Fre* represents the use of the space and frequency stream. From the Table 4, it can be seen that the frequency domain stream performs better than the spatial domain stream on clean images, but worse than the spatial domain stream on noisy images. This indicates that the frequency domain information is more suitable for estimating the quantisation step for clean images. However, for noisy images, the introduction of noise interferes with the distribution of the DCT. Therefore, frequency domain information performs poorly on noisy images. The proposed method fuses the spatial and frequency domain information through the SIDF module, which improves the robustness while taking into account the high accuracy. As a result, the accuracy of the two stream network is the highest on both clean and noisy images.

**The performance of classification and regression models**: As shown in Table 5, we demonstrate the performance of the classification and regression models at the quantization steps. Where, *Clas* represents classification models and *Reg* represents regression models. The methods in [2, 29] pointed out that for the estimation of discrete numbers, the commonly used approach involves employing the softmax function followed by the cross-entropy function,

which aids in the process of backpropagation. Quantisation steps are typically discrete numbers, so the classification model helps the network to dig out deeper compression traces into improving the accuracy. As shown in the Table 5, the average accuracy using the classification model is 6%-8% higher than regression.

**Table 5: Performance of classification and regression models.**

| $QF_1$ | UCID | | RAISE | | NRCS | |
|---|---|---|---|---|---|---|
| | *Reg* | *Clas* | *Reg* | *Clas* | *Reg* | *Clas* |
| 60 | 73.77 | **84.86** | 71.68 | **82.41** | 72.97 | **83.38** |
| 65 | 66.59 | **77.37** | 58.12 | **69.28** | 57.79 | **60.08** |
| 70 | 76.59 | **81.73** | 58.59 | **71.49** | 67.32 | **82.23** |
| 75 | 66.15 | **80.43** | 62.63 | **75.32** | 67.27 | **78.78** |
| 80 | 78.49 | **81.94** | 64.10 | **75.77** | 67.39 | **78.52** |
| 85 | 79.06 | **84.37** | 72.86 | **73.86** | 76.25 | **79.95** |
| 90 | 81.48 | **84.70** | 68.56 | **71.28** | 76.15 | **77.72** |
| 95 | 76.79 | **77.96** | 72.10 | **74.05** | 78.53 | **79.54** |
| Avg | 74.86 | **81.67** | 66.08 | **74.18** | 70.45 | **77.52** |

## 4.6 Conclusion

In this paper, we propose a two-branch network structure based on Swin Transformer. This two-branch network structure can extract the information in the spatial domain information with high robustness and frequency domain information with high accuracy, respectively. Moreover, we propose the SIDF-module to adaptively assign the weights of spatial domain features and frequency domain features. To further improve the performance of the algorithm on noisy images, we design a network with a classification structure to speed up the convergence of the network and improve the accuracy of the algorithm. Experiment results show that our method performs best on clean or noisy images.

**Acknowledgements**: This work was supported by the National Key R and D Program of China (Grants No. 2021QY0700), and National Natural Science Foundation of China (Grants No. 62072250, U23A20305, U23B2022,62371145, 62072480, 62172435, 62302249, 62272255, 62302248, U20B2065), and Zhongyuan Science and Technology Innovation Leading Talent Project of China (Grants No. 214200510019), and Open Foundation of Henan Key Laboratory of Cyberspace Situation Awareness (Grants No. HNTS2022002), and the Graduate Student Scientific Research Innovation Projects of Jiangsu Province (Grants No. KYCX24_1513).

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
