# OpenReview forum: "Advancing Quantization Steps Estimation : A Two-Stream Network Approach for Enhancing Robustness"
_acmmm.org/ACMMM/2024/Conference — MM2024 Poster_

### Official Review · Reviewer_tArZ · 2024-05-16

**Rating:** 3
**Confidence:** 3

**Summary:**

This paper proposes a two-stream network to robustly and accurately estimate the quantization step of JPEG images. The Swin Transformer is employed to build a two-branch network to extract compression traces of the image in both the frequency and spatial domains. The features from the two domains are then merged and classified to attain the quantization step. The experimental results show the average accuracy of the proposed method for both clean and noisy images.

**Strengths:**

1. The two-stream network takes advantage of the frequency domain features with deep-level characteristics and the spatial domain features with stronger robustness to allow highly accurate estimation.
2. Based on the attention mechanism, a Spatial-frequency Information Dynamic Fusion (SIDF) module is developed to adaptively fuse the features in two domains.
3. The network is constructed as a classification model in replace of a regression model.

**Limitations:**

1. There are some irregularities in the writing of the paper. In line 744, the grammar is incorrect. In Table 2, the results of the proposed method are bolded even where it performs worse. The conclusion is not presented as a separate section.
2. Data augmentation is applied during training, which improves the robustness. However, the comparative methods did not introduce data augmentation, so the performance improvement of the proposed method may be attributed to data augmentation.
3. There is no comparison with other methods in terms of computational complexity in the experiments.
4. There is no analysis of the differences between classification models and regression models.

**Suitability:**

2

---

### Official Review · Reviewer_viMB · 2024-05-19

**Rating:** 4
**Confidence:** 2

**Summary:**

This paper proposes a two-stream network approach for estimating the quantization step of JPEG images, aiming to improve robustness against various noise attacks. The network consists of two parallel streams: one for extracting spatial domain features and the other for frequency domain features. A SIDF module is introduced to dynamically assign weights to the two feature domains.

**Strengths:**

1. This paper introduces a novel network method for quantization step estimation in JPEG images, aimed at enhancing estimation robustness. This motivation is promising.

2. The two-stream network architecture extracts both spatial and frequency domain features, leveraging their respective strengths in robustness and accuracy.

**Limitations:**

1. In Section 1, this paper states that spatial domain features exhibit stronger robustness. However, the experiments detailed in Section 4.3 reveal that spatial domain methods do not demonstrate superior robustness when compared to frequency domain methods. This raise questions about the correctness of the initial statement.

2. In Section 1, the paper discusses how quantization step estimation can assist in identifying tampered or concealed information. However, it lacks experimental content related to image steganalysis for addressing these steganography methods, such as [a] and [b].

3. In Section 4.3, the selection of noise types utilized in the robustness experiments warrants further consideration. Including a broader range of noise types, such as Crop, Resize, Salt & Pepper, Style Transfer, and Superimpose Noise, could provide a more comprehensive evaluation of robustness.

4. In Section 4.5, the ablation study should not solely focus on accuracy evaluation but should also encompass robustness evaluation.

[a] Kaimeng Chen, Qingxiao Guan, Weiming Zhang, and Nenghai Yu. Reversible data hiding in encrypted images based on binary symmetric channel model and polar code. TDSC. 2023.
[a] Zijin Yang, Kejiang Chen, Kai Zeng, Weiming Zhang, and Nenghai Yu. 2024. Provably Secure Robust Image Steganography. TMM. 2024.

**Suitability:**

3

---

### Official Review · Reviewer_RUCK · 2024-05-21

**Rating:** 3
**Confidence:** 3

**Summary:**

This paper proposes a two-stream network classification model based on Swin Transformer architecture for estimating the quantization step of JPEG images. The proposal extracts features from both spatial and frequency domains and introduces a Spatial-frequency Information Dynamic Fusion (SIDF) module.

**Strengths:**

This paper presents a learned approach to quantization step estimation, which is a crucial aspect of image forensics. The two-stream network model can extract features from spatial and frequency domains, potentially leading to more accurate and robust estimations. The Spatial-frequency Information Dynamic Fusion (SIDF) module allows feature fusion. It has comprehensive discussions for evaluation, considering various noise conditions and comparing them with existing methods.

**Limitations:**

The novelty of the research method in this paper is insufficient and there are several problems.
The paper proposes a two-stream network with feature fusion, but feature fusion is a commonly used trick in the CV field.
In the ablation study,  the basic Swin Transformer block seems to bring more improvements than feature fusion. In addition, in Table 3 and Table 4, the experimental settings do not seem clear. Also, it seems that the accuracy of the proposed method (the accuracy of two blocks with SIDF module) cannot correspond to the above.
The paper does not discuss in detail the computational complexity of the proposed method, which could be a concern for practical applications.
It also does not provide a clear explanation of how the proposed method would perform on images of different sizes.
While the paper mentions the differences between classification and regression models, it does not delve into the specific details of the comparison.

**Suitability:**

2

---

### Official Review · Reviewer_texk · 2024-05-23

**Rating:** 5
**Confidence:** 3

**Summary:**

This paper presents a novel approach to JPEG image forensics, focusing on the estimation of quantization steps, which are critical for detecting image tampering and compression history. The authors propose a two-stream network structure utilizing the Swin Transformer to enhance robustness and accuracy. The spatial domain stream captures robust but less accurate features, while the frequency domain stream captures accurate but less robust features. They introduce a Spatial-frequency Information Dynamic Fusion (SIDF) module to dynamically fuse these features, and convert the network from a regression to a classification model to improve performance.

**Strengths:**

Innovative Network: The combination of spatial and frequency domain features using the Swin Transformer is a novel approach that leverages the strengths of both feature types to enhance the robustness and accuracy of quantization step estimation.
Comprehensive Experiments: The authors provide extensive experiments, demonstrating that their method achieves over 98% accuracy on clean images and maintains over 81% accuracy in robust environments.
The motivation for the model is clearly described: an analysis of the characteristics of existing models is provided, including the reasons for the low accuracy of spatial-domain features and the low robustness of frequency-domain features. The rationale for combining time-domain and frequency-domain features is well-explained.

**Limitations:**

1. Lack of interpretability in the model classification module: The authors design a neural network model with a classification structure instead of a regression structure to estimate the model's quantization step size, but they do not delve into the reasons for this design.
2. Complexity of the Model: The proposed two-stream network, while innovative, adds significant complexity compared to traditional methods. This might hinder its practical application.
3. End-to-End Training Complexity: The transition to an end-to-end classification model introduces additional training complexity. More discussion on the training process and computational requirements would be beneficial for readers considering practical implementations.
4. Limited Robustness Scenarios: Although the method shows robustness in some scenarios, the experiments do not cover all possible types of noise and distortions that can affect JPEG images. Further testing under a wider range of conditions would strengthen the claims of robustness.

**Suitability:**

2

---

### Meta-Review · Area_Chair_bSwn · 2024-06-30

**Recommendation:** Accept (Poster)
**Confidence:** 5

**Metareview:**

This paper received overall positive review comments. AC agrees with the reviewers to accept this paper. Congratulations! Please add the rebuttal contents to the final camera-ready version.